# An mRNA-based T-cell-inducing antigen strengthens COVID-19 vaccine against SARS-CoV-2 variants

Wanbo Tai[1,2,3,11], Shengyong Feng[2,11], Benjie Chai[2,11], Shuaiyao Lu [4,11], Guangyu Zhao[5,11], Dong Chen[6,7,11], Wenhai Yu[4,11], Liting Ren[8], Huicheng Shi[2], Jing Lu[9], Zhuming Cai[1,2], Mujia Pang[1], Xu Tan [8], Penghua Wang [10], Jinzhong Lin [9] ✉, Qiangming Sun [4] ✉, Xiaozhong Peng [4] ✉ & Gong Cheng [2] ✉

Herd immunity achieved through mass vaccination is an effective approach to prevent contagious diseases. Nonetheless, emerging SARS-CoV-2 variants with frequent mutations largely evaded humoral immunity induced by Spike-based COVID-19 vaccines. Herein, we develop a lipid nanoparticle (LNP)-formulated mRNA-based T-cell-inducing antigen, which targeted three SARS-CoV-2 proteome regions that enriched human HLA-I epitopes (HLA-EPs). Immunization of HLA-EPs induces potent cellular responses to prevent SARS-CoV-2 infection in humanized HLA-A*02:01/DR1 and HLA-A*11:01/DR1 transgenic mice. Of note, the sequences of HLA-EPs are highly conserved among SARS-CoV-2 variants of concern. In humanized HLA-transgenic mice and female rhesus macaques, dual immunization with the LNP-formulated mRNAs encoding HLA-EPs and the receptor-binding domain of the SARS-CoV-2 B.1.351 variant (RBD_{beta}) is more efficacious in preventing infection of SARS-CoV-2 Beta and Omicron BA.1 variants than single immunization of LNP-*RBD_{beta}*. This study demonstrates the necessity to strengthen the vaccine effectiveness by comprehensively stimulating both humoral and cellular responses, thereby offering insight for optimizing the design of COVID-19 vaccines.

Severe acute respiratory syndrome coronavirus-2 (SARS-CoV-2) is the etiological agent of the current pandemic coronavirus disease 2019 (COVID-19), which has killed over 6.8 million people worldwide[1]. Herd immunity achieved through mass vaccination is the most effective approach to preventing contagious diseases[2,3]. The currently authorized vaccines are based on the SARS-CoV-2 Spike (S) protein or its receptor-binding domain (RBD)[4,5], which can elicit potent neutralizing antibodies to block Spike-mediated viral entry into host cells[6]. Nonetheless, accumulating evidence has suggested that transient humoral responses are elicited by vaccination[7], while vaccination-induced neutralizing antibodies may rapidly wane over time[8]. In addition,

emerging SARS-CoV-2 variants with frequent mutations in the Spike protein may evade humoral immunity induced by vaccination or natural infection[9,10], and these mutations have accounted for many breakthrough infections in recent COVID-19 surges[11,12].

Ideal vaccines are expected to induce both protective humoral and cellular immunity, particularly long-lasting memory B/T-cell responses[13]. Indeed, a robust T-cell response provides potent immune protection against infection by many coronaviruses, including SARS-CoV-2[14–17]. Coronavirus-specific T cells are sufficient to effectively control viral pathogenesis even in the absence of neutralizing antibodies[18], suggesting that the generation of effective virus-specific

A full list of affiliations appears at the end of the paper. ✉e-mail: linjinzhong@fudan.edu.cn; qsun@imbcams.com.cn; pengxiaozhong@pumc.edu.cn; gongcheng@mail.tsinghua.edu.cn

cellular responses is essential for coronavirus clearance[19]. In addition, T-cell epitopes are comprehensively distributed throughout the whole SARS-CoV-2 proteome[20]. Since the epitopes recognized by neutralizing antibodies are concentrated in the mutation-prone RBD, emerging variants have been demonstrated to efficiently evade prior humoral immunity. However, the large repertoire of SARS-CoV-2-reactive T cells is likely still protective against emerging variants[21,22]. Therefore, the induction of broadly protective cellular responses should be a feasible approach to strengthen the effectiveness of COVID-19 vaccines. In this study, we developed an mRNA-based, T-cell-inducing antigen that encodes 3 SARS-CoV-2 peptides enriching human HLA-I epitopes, thus enabling the induction of broad and potent cellular responses. Dual immunization with both these T-cell-inducing and RBD-based mRNA antigens showed more efficacious in preventing SARS-CoV-2 infection than RBD vaccination alone in both humanized HLA-transgenic mice and nonhuman primates. These results provide strong evidence for the necessity of dual immunization to comprehensively stimulate both humoral and cellular responses for controlling COVID-19 pandemic.

## Results

### Identification of human HLA-I epitope-enriched regions in the SARS-CoV-2 proteome

The class I human leukocyte antigen (HLA-I) complex shows high polymorphism, resulting in the broad diversity of $CD8^+$ T-lymphocyte epitopes among human individuals[23]. We aimed to identify regions with diverse HLA-I-specific epitopes throughout SARS-CoV-2 open reading frames (ORFs), thus enabling the development of a recombinant antigen that induces broad and potent virus-specific responses by cytotoxic T lymphocytes (CTLs) (Fig. 1a). To address this, we predicted the epitopes for 78 HLA-I alleles, which are the most common in the human population (Supplementary Table 1), in the SARS-CoV-2 proteome with both the NetMHCpan and IEDB bioinformatic tools[24–27]. The predicted HLA-I epitopes with an affinity value ($IC_{50}$) less than 10 nM were defined as effective epitopes. Thus, four fragments with more than 20 predicted effective epitopes per 100 amino acids (nonstructural protein (NSP)-$3_{1443–1605}$, NSP-$4_{232–444}$, NSP-$6_{1–201}$, and Membrane (M)$_{1–113}$) were selected as HLA-I epitope-enriched peptides for further investigation (Fig. 1b, c). Nonetheless, several studies have identified $CD8^+$ T-cell epitopes in the N protein of SARS-CoV-2[28–30]. Although 12 HLA-I epitopes were identified in SARS-CoV-2 nucleocapsid (N) protein by the defined criteria of bioinformatic predictions (Fig. 1b, c and Supplementary Data 1), We still chose the $N_{Full-length}$ as a potential HLA-I epitope-enriched peptide. Three peptides with a low number of predicted HLA-I epitopes, including NSP-$1_{1–180}$, NSP-$3_{1066–1278}$ and NSP-$14_{330–490}$, were selected as negative controls (Fig. 1b, c). We next assessed whether ectopic expression of these peptides could activate CTLs from convalescent COVID-19 patients through a reporter cell-based epitope expression system[26,31]. Both an HLA subtype (HLA-A*02:01 or HLA-A*11:01) and an HLA-I epitope-enriched or a negative control peptide from SARS-CoV-2 proteome were ectopically expressed in human 293 reporter cells. In addition, the reporter cells also expressed a modified infrared fluorescence protein (IFP) containing a granzyme B (GzB) cleavage sequence resulting in a fluorescence signal appearing immediately after cleavage by GzB[32]. Thus, the reporter cells, in which the SARS-CoV-2 epitopes were presented by a specific HLA-I subtype, were incubated with $CD8^+$ T cells isolated from the convalescent COVID-19 patients (Supplementary Table 2). The reporter cells presenting the epitopes activated cognate $CD8^+$ T cells, which secreted GzB which induced apoptosis in the reporter cells. Subsequently, the modified IFP in the reporter cells was cleaved to produce the fluorescence signal (Fig. 1a). Of note, ectopic expression of 3 peptides located in nonstructural protein (NSP) $-3_{1443–1605}$, NSP-$4_{232–444}$ and NSP-$6_{1–201}$ stimulated higher fluorescence in all tested reporter cells expressing a specific HLA-I subtype

(Fig. 1d, e). In contrast, ectopic expression of either the $M_{1–113}$ or $N_{Full-length}$ protein was unable to robustly activate $CD8^+$ T cells from the convalescent COVID-19 patients (Fig. 1d, e). Overall, we identified 3 specific HLA-I epitope-enriched peptides in the SARS-CoV-2 non-structural proteins that were highly immunogenic and thus activated human CTL responses.

### Assessment of the immunogenicity of HLA-I epitope-enriched peptides in humanized HLA-transgenic mice

We next investigated the immunogenicity of identified HLA-I epitope-enriched antigens that activate CTL responses against SARS-CoV-2 in animals. A genetic mRNA format allows for in situ production of epitope-enriched peptides, which could efficiently access the HLA compartment of antigen-presenting cells[33]. Therefore, we generated a recombinant mRNA transcript encoding 3 HLA-I epitope-enriched peptides (HLA-EPs), representing the NSP-$3_{1443–1605}$, NSP-$4_{232–444}$, and NSP-$6_{1–201}$ regions in tandem. Three peptides with a few HLA-I epitopes, including NSP-$1_{1–180}$, NSP-$3_{1066–1278}$ and NSP-$14_{330–490}$, were exploited to construct a recombinant mRNA (HLA-NC) as a negative control. A sequence encoding ubiquitin (Ub) with a G76A amino acid modification was placed immediately upstream of the *HLA-EPs* and *HLA-NC* sequences. An A-A-Y spacer was inserted between the three peptides in either HLA-EPs or HLA-NC[34] (Fig. 2a). The mRNAs were encapsulated with lipid nanoparticle (LNP)-formulated nucleoside-modified mRNA technology (LNP-*HLA-EPs* and LNP-*NC*). The physical features and tissue distribution of the LNPs were characterized with a luciferase mRNA sequence (Supplementary Fig. 1). Based on this design, ubiquitinated antigens could be immediately degraded in the proteasome for HLA presentation[35]. The efficiency of HLA-EPs degradation was determined by flow cytometry in mRNA-transfected HEK293T cells with or without treatment with the proteasome inhibitor MG132 (Fig. 2b, c).

Subsequently, we employed two humanized H-2 class I-/class II-knockout C57BL/6 J strains that express either human HLA-A*02:01/DR1 or HLA-A*11:01/DR1, to evaluate the immunogenicity of LNP-*HLA-EPs* in vivo[36,37]. Either LNP-*HLA-EPs* (0.5 μg/dose or 10 μg/dose) or LNP-*NC* (10 μg/dose) was inoculated into both strains of HLA-transgenic mice following a full vaccination procedure (two doses with a 3-week interval). Twenty-one days after receiving the booster vaccine, the LNP-*HLA-EPs*-immunized HLA-transgenic mice displayed a significant increase in the frequencies of $CD8^+$ but not $CD4^+$ T cells to total T lymphocytes, compared to that of LNP-*NC* (Supplementary Fig. 2a, b). Antigen-specific memory immune responses provide long-term protection[38]. Notably, immunization significantly enlarged the population of $CD44^+CD62L^-CD8^+$ effector memory T cells (Tem) and $CD44^+CD62L^+CD8^+$ central memory T cells (Tcm) in both strains of HLA-transgenic mice (Supplementary Fig. 2c, d), suggesting HLA-EPs-mediated activation of cellular immunity. We next examined the epitope-specific responses of cytotoxic lymphocytes in these immunized mice. The proportion of epitope-specific splenocytes producing interferon-gamma (IFN-γ) following stimulation with the HLA-EPs peptide pool was dramatically enhanced in the LNP-*HLA-EPs*-immunized mice compared to control mice (Fig. 2d–f). The increases in different aspects of the $CD8^+$ T-cell response were dose-dependent.

Previous studies have indicated that virus-specific T cells provide potent protection against many coronaviruses[17,39]. We next assessed the protective efficacy of LNP-*HLA-EPs* against SARS-CoV-2 infection in HLA-transgenic mice by $CD8^+$ T cells depletion and live SARS-CoV-2 challenge. Both HLA-A*02:01/DR1 and HLA-A*11:01/DR1 transgenic mice were inoculated with either a low (0.5 μg) or a high (10 μg) doses of LNP-*HLA-EPs* or LNP-*NC* (10 μg). Twenty days after the booster immunization, the mice under all vaccination schemes were randomly divided into two subgroups and then inoculated twice with either an anti-CD8-α antibody or an isotype IgG2b control. On the first day post-inoculation with the anti-CD8-α antibody, $CD8^+$ T cells were fully depleted from the peripheral blood (Supplementary Fig. 2a). The mice

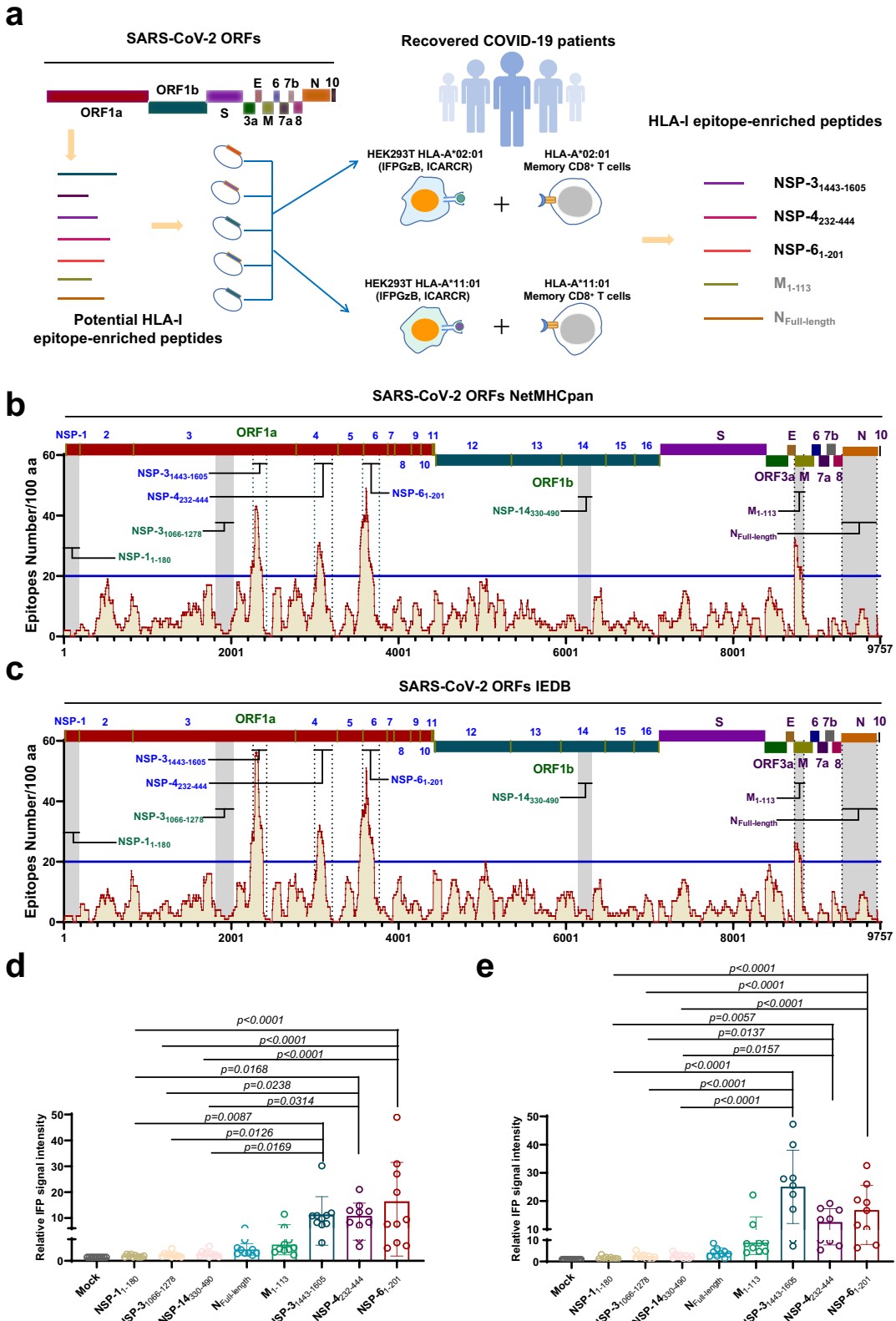

that received the isotype control antibody showed a normal fraction of CD8+ T cells in the peripheral blood. The proportions of CD8+ T cells in both the peripheral blood and spleens of LNP-*HLA-EPs*-immunized animals was higher than that in the LNP-*NC*-immunized mice (Supplementary Fig. 2a, b), suggesting activation of T-cell responses by the LNP-*HLA-EPs*-immunization. Subsequently, the immunized HLA-transgenic mice were challenged with $1.4 \times 10^5$ PFU of SARS-CoV-2

B.1.351 (Beta) variant via nasal instillation. The viral loads in the lungs were quantitated by a plaque assay and qRT-PCR at 4 days postinfection, and histopathological damage was assessed by haematoxylin−eosin (H&E) staining. Intriguingly, immunization with LNP-*HLA-EPs* significantly reduced the SARS-CoV-2 loads (Fig. 2g, h and Supplementary Fig. 9a) and pathological damage (Fig. 2i, j) in the lungs of both HLA-A*02:01/DR1 (left panel) and HLA-A*11:01/DR1 (right

**Fig. 1 | Identification of HLA-I epitope-enriched peptides that activate CD8+ T lymphocytes from convalescent COVID-19 patients. a** Schematic diagram of experimental design. The epitopes for 78 HLA-I alleles were predicted by both the NetMHCpan and IEDB bioinformatic tools. The peptides enriching HLA-I epitopes were ectopically expressed to assess the capability to activate CD8+ T lymphocytes from convalescent COVID-19 patients through a reporter cell-based epitope expression system. **b, c** Identification of the fragments enriching human cytotoxic T-lymphocyte epitopes in the SARS-CoV-2 proteome. The distribution of potential cytotoxic HLA-I epitopes was predicted by in silico analyses of NetMHCpan (**b**) and IEDB (**c**). Four regions, NSP-3$_{1443-1605}$, NSP-4$_{232-444}$, NSP-6$_{1-201}$, and M$_{1-113}$, with more than 20 effective epitopes (affinity concentration >10 nM) per 100 amino acids were selected as HLA-I epitope-enriched peptides for further investigation, and the three regions, NSP-1$_{1-180}$, NSP-3$_{1066-1278}$, and NSP-14$_{330-490}$ with fewest predicted effective epitopes were set as controls. The N protein was included in further investigation. The horizontal lines denote the threshold for selecting the HLA-I epitope enrichment regions. **d, e** Activation of CD8+ T lymphocytes from

convalescent COVID-19 patients. HEK293T cells were transfected to stably express the pHAGE_EF1a_ICADCR, pHAGE_EF1a_IFPGZB, and pHAGE_EF1aHLA vectors. Either HLA-A*02:01 (**d**) or HLA-A*11:01 (**e**) overexpressing HEK293T cells were transfected with the genes encoding the NSP-1$_{1-180}$, NSP-3$_{1066-1278}$, NSP-1$_{4330-490}$, N$_{Full-length}$, M$_{1-113}$, NSP-3$_{1443-1605}$, NSP-4$_{232-444}$, or NSP-6$_{1-201}$, respectively. At 36 h post-transfection, $1 \times 10^6$ HEK293T cells were co-cultured with $2 \times 10^5$ HLA genotype-matched memory CD8+ T cells from convalescent COVID-19 patients ($n = 10$ for HLA-A*02:01, and $n = 9$ for HLA-A*11:01) for 12 h, and activation of CD8+ T cells was characterized by the relative IFP signal intensity in the co-cultred HEK293T cells using the flow cytometric measurement. All data are presented as the mean ± S.E.M from the indicated number of biological repeats. Source data are provided as a Source Data file. Experiments were repeated twice independently with similar results. Both independent experiments contain two technical repeats (**d, e**). Adjusted $p$ values were determined by one-way ANOVA with Tukey's multiple comparison post-hoc two-sided tests.

panel) transgenic mice in a dose-dependent manner. Nonetheless, this protective efficacy of LNP-*HLA-EPs* was largely abolished in CD8+ T-cell-depleted mice, indicating that the anti-SARS-CoV-2 response elicited by the LNP-*HLA-EPs* was CD8+ T-cell-dependent (Fig. 2g–j).

Emerging SARS-CoV-2 variants significantly compromise the efficacy of current Spike-based vaccines because of continuous mutations in the epitopes recognized by neutralizing antibodies[40]. The sequences of three HLA-I epitope-enriched peptides were found to be highly conserved among the SARS-CoV-2 variants of concern (VOCs). Nonetheless, several substitutions were shown to exist in the SARS-CoV-2 NSP-4$_{232-444}$ and NSP-6$_{1-201}$ peptides, and the substitutions did not influence the immunogenicity of epitopes predicted by bioinformatic analyses (Supplementary Table 3). We assessed whether these substitutions influence the SARS-CoV-2-specific CD8+ T-cell response with an HLA class I tetramer assay. Of note, most substitutions in these peptides were not located in the effective epitopes predicted by the defined criteria (Fig. 1b, c). The HLA-A*02:01 or HLA-A*11:01-specific epitopes containing the substitutions of the SARS-CoV-2 VOCs were synthesized (Supplementary Table 4). The corresponding epitopes of the wild-type strain served as controls. We next constructed HLA-A*02:01 and HLA-A*11:01 HLA-I tetramers with the epitopes and used them to stain CD8+ T cells from LNP-*HLA-EPs*-immunized transgenic mice. There was no significant difference in the tetramer-positive faction between the original and mutated epitopes (Supplementary Fig. 3), suggesting that the immunogenicity of HLA-EPs may not be influenced by the mutations in SARS-CoV-2 variants. Thus, immunization with HLA-EPs may induce broadly protective cellular immune responses against the known SARS-CoV-2 VOCs, including the recently emerged Omicron variants.

**Dual immunization with HLA-EPs and SARS-CoV-2 RBD antigens protects humanized HLA-transgenic mice from infection with SARS-CoV-2 variants**

Emerging SARS-CoV-2 VOCs frequently escape prior humoral immunity, thereby largely reducing the efficacy of current COVID-19 vaccines[9]. Given that the immunogenicity of HLA-EPs was not impaired by the mutations in the SARS-CoV-2 VOCs, we proposed that dual immunization with HLA-EPs and a SARS-CoV-2 RBD antigen might be much more effective against the currently circulating SARS-CoV-2 variants. The SARS-CoV-2 B.1.351 variant has a strong ability to escape neutralizing immunity induced by vaccination or natural infection[41,42]. We, therefore, chose the RBD of the SARS-CoV-2 B.1.351 strain to generate an mRNA-based antigen in an LNP-formulated system (defined as LNP-*RBD*$_{beta}$) (Supplementary Fig. 4a). The expression of LNP-*RBD*$_{beta}$ was validated by both Western blotting and flow cytometry (Supplementary Fig. 4b, c). Next, we exploited the HLA-A*02:01/DR1 transgenic mice as an experimental model to assess the efficacy of dual

immunization in protection from SARS-CoV-2 infection. The HLA-transgenic mice were immunized with a combination of 0.5 μg LNP-*RBD*$_{beta}$ and 0.5 μg LNP-*HLA-EPs*. Mice immunized with either 0.5 μg LNP-*RBD*$_{beta}$ only or PBS served as controls. The animals were given two doses of the dual antigens with a 3-week interval (Fig. 3a). We first examined the signaling axis of C-X-C chemokine ligand 13 (CXCL13), which guides dynamic germinal center reactions and B-cell selection[43,44]. Indeed, the expression of CXCL13 and its receptor CXCR5 was significantly increased in the lymph nodes of the animals immunized with LNP-*RBD*$_{beta}$ and LNP-*RBD*$_{beta}$ + LNP-*HLA-EPs* (Supplementary Fig. 5a, b). Consistently, the frequencies of germinal center B (GC B) cells (B220+CD95+GL-7+) and T follicular helper (Tfh) cells (CD4+CD185+PD-1+) were higher in the mice that received LNP-*RBD*$_{beta}$ alone or the dual vaccines than in PBS control mice (Fig. 3b). Twenty-one days after the booster dose, the RBD-specific IgG titers in the serum of immunized mice were measured by an enzyme-linked immunosorbent assay (ELISA). Antisera from the LNP-*RBD*$_{beta}$-immunized animals broadly recognized the RBDs of multiple SARS-CoV-2 circulating strains, including Wuhan-Hu-1 (wild-type), B.1.1.7 (Alpha), B.1.351 (Beta), P.1 (Gamma), B.1.617.2 (Delta) and B.1.1.529 (Omicron BA.1). There was no significant difference in RBD-specific IgG titers between LNP-*RBD*$_{beta}$- and LNP-*RBD*$_{beta}$ + LNP-*HLA-EPs*-immunized animals (Fig. 3c). Nonetheless, the specificity of anti-RBD$_{beta}$ serum for the RBD of B.1.1.529 (Omicron BA.1) was significantly lower than that for RBD$_{beta}$ and the RBDs of other variants (Fig. 3c). We next investigated the neutralization of the wild-type or variant SARS-CoV-2 strains by the serum antibodies in immunized animals. Of note, although sera from both LNP-*RBD*$_{beta}$- and LNP-*RBD*$_{beta}$ + LNP-*HLA-EPs*-vaccinated animals were capable of neutralizing several SARS-CoV-2 variants, the neutralizing antibody titers against the Omicron BA.1 variant were significantly lower than those against the other strains (Fig. 3d). These data suggest that LNP-*RBD*$_{beta}$-induced antibodies showed neutralizing activity against some SARS-CoV-2 variants. We next assessed the cellular immune responses in mice immunized with both LNP-*HLA-EPs* and LNP-*RBD*$_{beta}$ mRNA antigens. Splenocytes were isolated from either immunized or control-treated HLA-transgenic mice at 21 days after the booster dose. The LNP-*HLA-EPs* + LNP-*RBD*$_{beta}$-immunized HLA-transgenic animals showed a significant enhancement in the ratios of CD8+ T cells and CD4+ T cells to total T lymphocytes (Supplementary Fig. 6a, b). Subsequently, we assessed the epitope-specific responses of cytotoxic lymphocytes in these immunized mice. After stimulation with the peptide pool of HLA-EPs and RBD$_{beta}$, the number of epitope-specific splenocytes producing interferon-gamma (IFN-γ) was dramatically enhanced in the dual-immunized mice, as measured with an ELISpot assay (Fig. 3e), but singly and dually

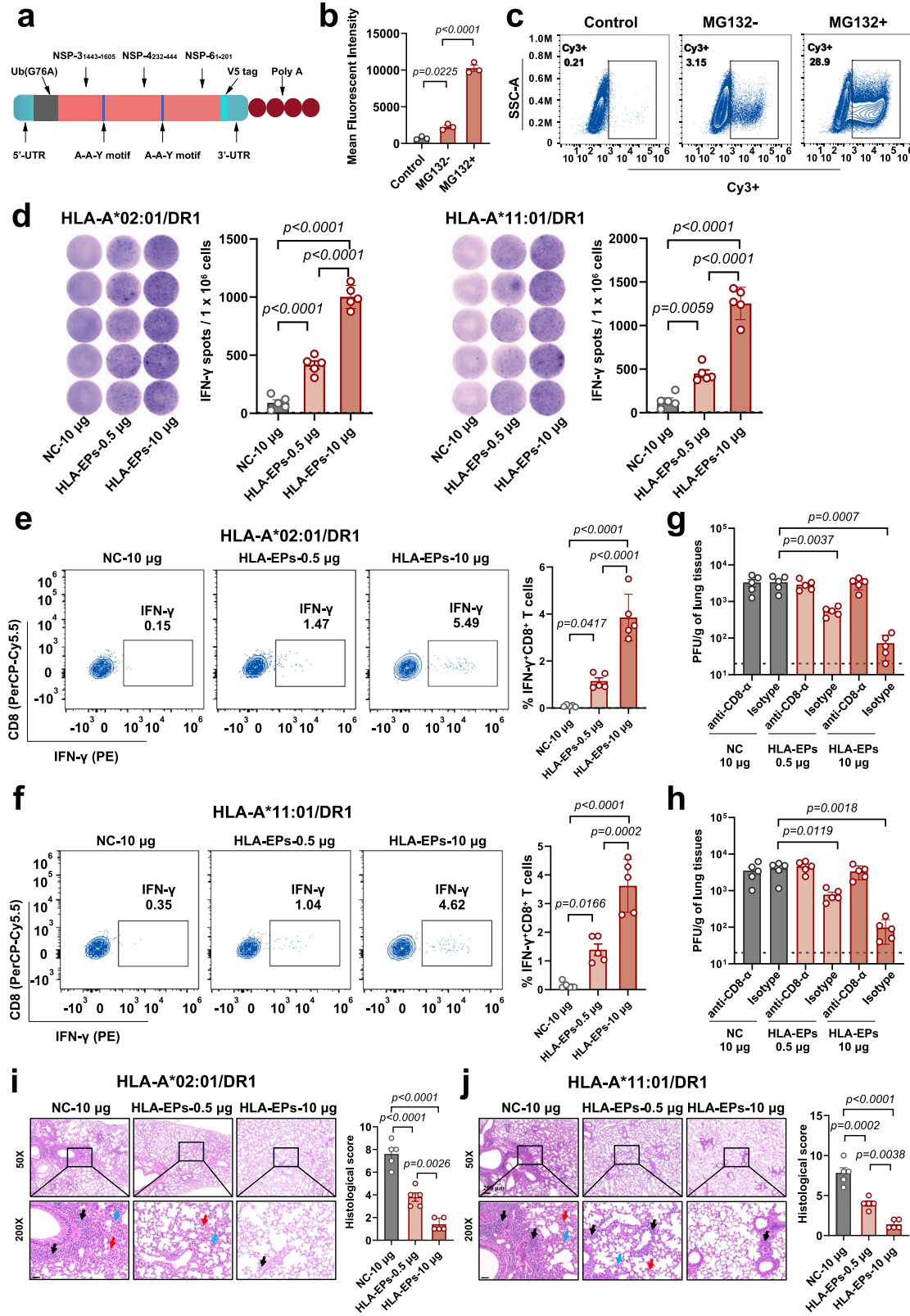

immunized mice were comparable in the number of interleukin-4 (IL-4) producing splenocytes (Fig. 3f), indicating a Th1-biased response elicited by HLA-EPs. In addition, the frequencies of the epitope-specific splenocytes from the dually immunized mice were capable of releasing the cytokines IFN-γ and tumor necrosis factor-alpha (TNF-α) were higher than those from LNP-*RBD*$_{beta}$- immunized or PBS inoculated transgenic mice, as determined by

intracellular cytokine staining (Fig. 3g, h). These results indicate that dual immunization with HLA-EPs and a SARS-CoV-2 RBD antigen induces robust antigen-specific immune responses in HLA-transgenic animals.

We further explored the protective efficacy of this dual immunization against SARS-CoV-2 infection. Immunized mice were challenged with the SARS-CoV-2 B.1.351 variant ($1.4 \times 10^5$ PFU/mouse) at 21 days

**Fig. 2 | Immunogenicity of HLA-EPs in both HLA-A*02:01/DR1 and HLA-A*11:01/DR1 transgenic mice. a** Schematic diagram of the *HLA-EPs* mRNA construct. **b, c** The expression of HLA-EPs in HEK293T cells. The results are shown by the mean fluorescent intensity (MFI) of HLA-EPs positive cells (**b**) and the representative flow cytometry plots (**c**) of 3 biological replicates containing 3 technical replicates. **d–f** The proliferation of epitope-specific splenocytes producing IFN-γ following stimulation with the peptide pool. The proliferation of epitope-specific, IFN-γ producing splenocytes was measured by ELISpot. Each point represents the mean of 3 technical replicates, with a limit of detection (LOD) = 2, and representative dots are shown (**d**). The frequency of IFN-γ positive CD8⁺ T cells was measured by an ICS (**e, f**). The representative flow cytometry plots are presented, and symbols represent individual mice. **g–j** The CD8⁺ T-cell-dependent protection of LNP-*HLA-EPs* against SARS-CoV-2. Both vaccinated HLA-A*02:01/DR1 and HLA-A*11:01/DR1 transgenic mice with or without CD8⁺ T-cell depletion, were infected with the SARS-CoV-2 B.1.351 variant (1.4 × 10⁵ PFU/mouse). Viral titers in lung tissues of HLA-A*02:01/DR1 and HLA-A*11:01/DR1 transgenic mice, with and without CD8⁺ T-cell depletion after 4 days post-inoculation (n = 5), were determined by a plaque assay with two technical replicates (**g, h**). The LOD was 10 PFU per gram of tissue. Histopathological changes in the lungs of challenged mice were evaluated by an H&E staining (**i, j**). Images derived from one representative animal in each group with sites of inflammatory cell infiltration (black arrows), blood clots (blue arrows), and alveolar deformation (red arrows) are presented. Scale bar is 50 μm. All data (**b, d–j**) are presented as the mean ± S.E.M. from the indicated number of biological repeats, and statistical significance was calculated via one-way ANOVA with Tukey's multiple comparison post hoc two-sided tests (**b, d–f, i, j**) or two-way ANOVA with Bonferroni's multiple comparisons (**g, h**). *P* values were adjusted for multiple comparisons. Source data are provided as a Source Data file. Data are representative of one (**i, j**), two (**d–h**), or three (**a, b**) independent experiments with similar results.

after the booster vaccination. Viral loads in the lungs and tracheae were assessed at 4 days after the viral challenge. Compared to mock and single immunization, the dual immunization regimen further prevented SARS-CoV-2 infection in HLA-A*02:01/DR1 transgenic mice (Fig. 3i and Supplementary Fig. 9b). Histological examination at 4 days post challenge revealed a typical pathology of viral interstitial pneumonia, such as extensive inflammatory cell infiltration, exudative pleural effusion and proliferative alveolar epithelium in the control mice (Fig. 3j, upper panel). Mild pathological changes were still present in the mice immunized with LNP-*RBD_beta* alone (Fig. 3j, upper panel). Of note, the mice dually immunized with both LNP-*RBD_beta* and LNP-*HLA-EPs* only showed very light signs of lung pathology (Fig. 3j).

The aforementioned results demonstrated that the immunogenicity of HLA-EPs was not influenced by the mutations in SARS-CoV-2 VOCs (Supplementary Fig. 3). We, therefore, assessed the protective efficacy of dual immunization with LNP-*RBD_beta* and LNP-*HLA-EPs* against the Omicron BA.1 variant in the HLA-A*02:01/DR1 transgenic mice. Immunized mice were challenged with 2.0 × 10⁵ PFU SARS-CoV-2 B.1.1.529 variant via nasal instillation at 21 days after the booster dose. The SARS-CoV-2 loads in the lungs and tracheae were measured at 4 days postinfection, via both a plaque assay and qRT-PCR assays, and the pathological damages in the lungs were evaluated. LNP-*RBD_beta* failed to render mice protection against the B.1.1.529 variant in terms of viral loads, while the infectious virus was hardly detectable in mice immunized with LNP-*RBD_beta* and LNP-*HLA-EPs* (Fig. 3i and Supplementary Fig. 9b). Assessment of pathological damages in lung tissues indicated that no significant pathological feature was observed in the lungs of dually immunized mice, while an immune infiltration was observed in the alveolar space of mice immunized with LNP-*RBD_beta* alone or the placebo control (Fig. 3j, lower panel), suggesting that dual immunization with HLA-EPs and SARS-CoV-2 RBD_beta provided potent protection from infection with SARS-CoV-2 variants. Notably, depletion of CD8⁺ T cells by inoculation of an anti-CD8-α antibody abolished the protection against either SARS-CoV-2 B.1.351 or B.1.1.529 infection in LNP-*HLA-EPs* + LNP-*RBD_beta*-immunized transgenic mice (Fig. 3i and Supplementary Fig. 9b), further verifying that the superiority in the protection of the dual vaccine design was dependent on cellular immunity. Overall, in the mice immunized by a low mRNA-LNP dose, the dual vaccination showed much more effective protection than that of single RBD_beta immunization.

### Dual immunization with HLA-EPs and a SARS-CoV-2 RBD produces optimal protection against SARS-CoV-2 in nonhuman primates

We next evaluated the immunogenicity and protective efficacy of dual immunization with LNP-*HLA-EPs* and LNP-*RBD_beta* in rhesus macaques (*Macaca mulatta*). Macaques were intramuscularly immunized with two doses of 100 μg LNP-*RBD_beta* and 100 μg LNP-*HLA-EPs* with a 3-week interval (Fig. 4a). The groups of macaques immunized with 100 μg LNP-*RBD_beta* alone or PBS served as controls. RBD-binding IgGs were readily detectable by 14 days after the first dose, and the levels were further increased 14 days after the second dose (Fig. 4b). We next assessed whether the serum antibodies in immunized animals are capable of neutralizing these SARS-CoV-2 VOCs. Broadly neutralizing antibodies presenting after the booster dose was assessed with pseudoviruses (Fig. 4c) or infectious SARS-CoV-2 VOCs (Fig. 4d). Of note, the neutralizing antibody titers against the Omicron BA.1 variant in the macaque serum were significantly lower than those against other strains (Fig. 4c, d). We further investigated the antigen-specific T-cell responses in macaques immunized with LNP-*RBD_beta* alone or LNP-*HLA-EPs* + LNP-*RBD_beta*. The PBMCs were isolated from immunized and control monkeys at 14 days post-booster immunization, and subsequently re-stimulated with the peptides used for immunization. The numbers of IFN-γ-producing T cells in the dually immunized macaques were significantly higher than that in the LNP-*RBD_beta*-immunized and control animals (Fig. 4e and Supplementary Fig. 7a). However, the numbers of IL-4-producing T cells were the same among all the experimental groups (Fig. 4f and Supplementary Fig. 7a). The expression of IFN-γ and TNF-α but not that of IL-4 and IL-5 was also upregulated in the dually immunized macaques (Supplementary Fig. 7b), thereby indicating a Th1-biased response elicited by HLA-EPs in these nonhuman primates (Fig. 4e, f and Supplementary Fig. 7).

The immunized macaques were then challenged with 7.0 × 10⁵ PFU of SARS-CoV-2 (B.1.351 variant) by the intranasal and intratracheal route[45]. The body temperatures and weights of all animals fluctuated within the normal range after a viral challenge from 0 to 7 days post-inoculation (Supplementary Fig. 8a). The viral loads of throat and anal swabs were much lower in the LNP-*HLA-EPs* + LNP-*RBD_beta*-immunized animals than in the control macaques during the whole evaluation period after viral challenge (Fig. 4g). Of note, although the viral sgRNA loads of these swabs taken from dually immunized animals were low and detectable in the early days of infection, they were completely undetectable by 3 or 5 days post-infection. Subsequently, the viral loads in multiple sites of infected macaque lungs were evaluated at 7 days postinfection by qRT-PCR assays (Fig. 4h). Compared to control treatment, immunization with LNP-*RBD_beta* alleviated SARS-CoV-2 infection in all lung lobes (Fig. 4h). The lung tissues of both control and LNP-*RBD_beta*-immunized monkeys showed typical interstitial pneumonia, which is a key feature of COVID-19[46], such as apparent thickening of the alveolar walls, loss of recognizable architecture, diffuse hemorrhage, and massive immune infiltrates, via an H&E staining (Supplementary Fig. 8b). In contrast, dual immunization with LNP-*HLA-EPs* and LNP-*RBD_beta* completely prevented SARS-CoV-2 infection at all test lobes except the right-up one (Fig. 4h). Consistently, the lung tissues of the dually immunized animals did not show any significant histopathological damage (Supplementary Fig. 8b). Overall, the dual vaccination strategy effectively prevented SARS-CoV-2 infection in nonhuman primates.

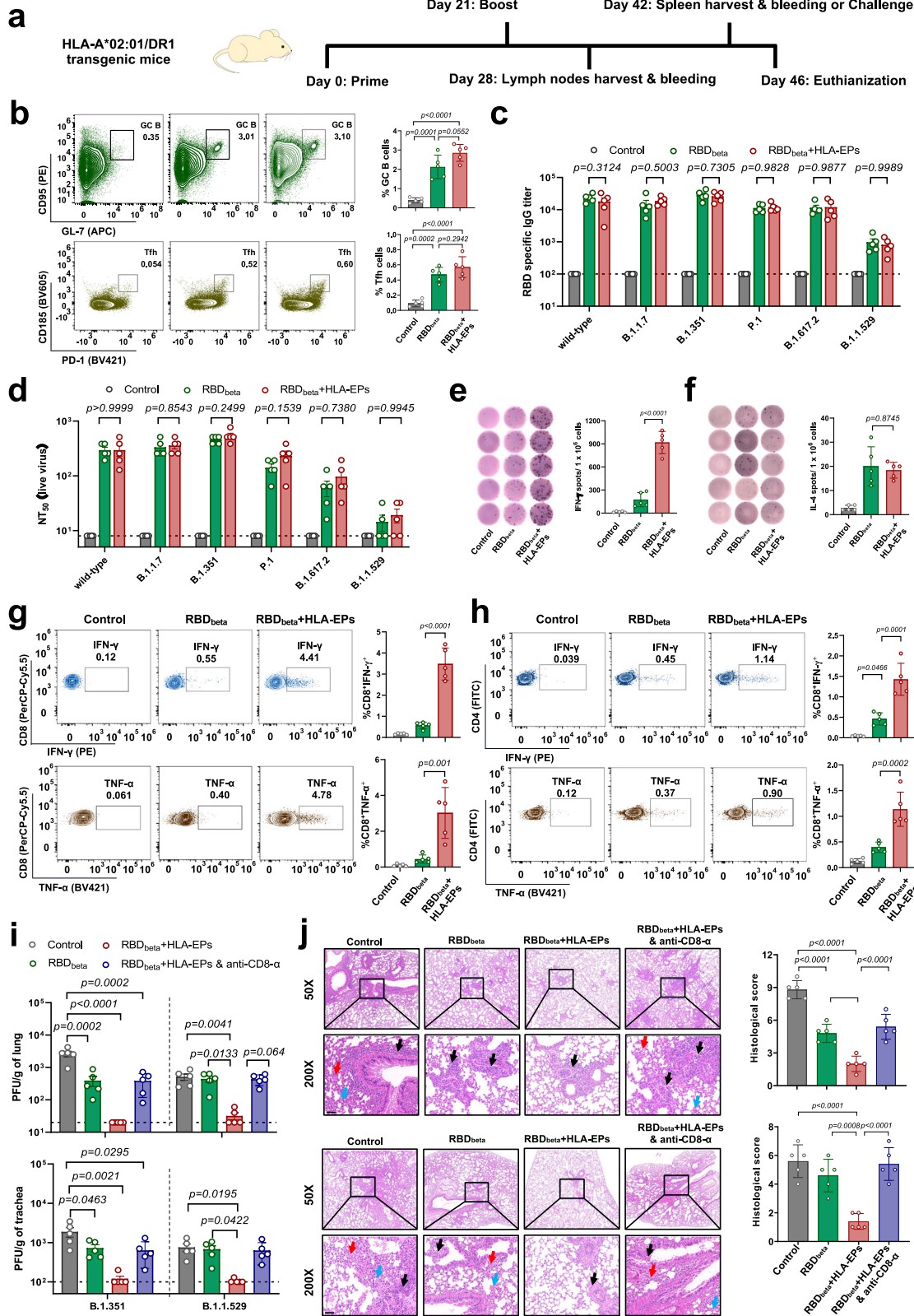

## Discussion

Current COVID-19 vaccines induce neutralizing antibodies against the SARS-CoV-2 Spike protein; however, their efficacy against emerging variants, particularly the Omicron lineages, is significantly reduced. In addition to neutralizing antibodies, robust and broad cellular responses provide potent immune protection against COVID-19[45,46]. The abundance of T lymphocytes has been closely related to clinical manifestations in COVID-19 patients, thus serving as a clinical predictor of a poor prognosis for COVID-19[47]. Coronavirus-specific cellular responses were detected in most infected patients, including asymptomatic patients with undetectable antibody responses[22], suggesting that coronavirus-specific T cells could be sufficient to clear viruses in the absence of neutralizing antibodies. Some asymptomatic or mild COVID-19 patients have exhibited cellular immune responses without

**Fig. 3 | Protection of HLA-A*02:01/DR transgenic mice from SARS-CoV-2 infection by dual immunization with SARS-CoV-2 HLA-EPs and RBD antigens.** **a** Schematic diagram of the immunization, sample collection, and challenge schedule. **b–d** The humoral immune activation and antibody production after immunization. The lymph nodes collected (*n* = 5) were analyzed for GC B cells and Tfh cells (**b**). SARS-CoV RBD-specific IgG antibodies in the sera collected 21 days after the booster vaccination were detected with ELISA (**c**). The neutralizing antibodies against SARS-CoV-2 variants were detected with live viruses-based neutralization assays (**d**). The dashed lines represent that the LOD was 100 for ELISA, and that the lower limit of quantification was 8 for live virus neutralization. **e–h** The cellular immune response. Splenocytes of immunized HLA-A*02:01/DR1 transgenic mice (*n* = 5) collected were restimulated ex vivo and subjected to IFN-γ ELISpot (**e**) (LOD = 5) and IL-4 ELISpot (**f**) (LOD = 2). The production of IFN-γ and TNF-α by CD8$^+$ T cells (**g**) and CD4$^+$ T cells was analyzed by an ICS (**h**). **i, j** The protective efficacy of the vaccine against the SARS-CoV-2 B.1.351 (Beta) and B.1.1.529 (Omicron BA.1)

variants. Vaccinated HLA-A*02:01/DR1 transgenic mice (*n* = 5) were challenged with the SARS-CoV-2 B.1.351 variant or B.1.1.529 variant, and viral titers at 4 days post-inoculation in the lung (**i**-upper panel) and trachea (**i**-bottom panel) tissues were determined by a plaque assay, with LODs being 20 and 100. The histopathological changes of the lungs at 4 days post-inoculation by the B.1.351 (**j**-upper panel) or B.1.1.529 (**j**-bottom panel) variant were evaluated after H&E staining. Images derived from one representative animal in each group with sites of inflammatory cell infiltration (black arrows), blood clots (blue arrows), and alveolar deformation (red arrows) are presented. Scale bar is 50 μm. Data (**b**–**j**) are shown as the mean ± S.E.M from individual mice. Source data are provided as a Source Data file. One (**b, j**) or two (**c**–**i**) independent experiments were performed with 2 technical replicates. Adjusted *p* values for statistical analysis were calculated via one-way ANOVA with Tukey's multiple comparison post-hoc two-sided tests (**b, e**–**j**) or two-way ANOVA with Bonferroni's multiple comparisons (**c, d**).

seroconversion[48,49]. In addition, several studies have shown that although some SARS-CoV-2 variants, in particular the Omicron lineages, are resistant to the neutralizing antibodies elicited by current vaccines or prior infection, they are still universally sensitive to the CD8$^+$ T-cell response induced by current vaccines or prior infection[50,51]. Therefore, accumulating evidence suggests an essential role for virus-specific T-cell responses in the prevention of COVID-19. In this study, we first developed an LNP-formulated mRNA antigen encoding a peptide enriched in human HLA-I epitopes, named HLA-EPs, which induced strong T-cell immunity. Immunization with LNP-*HLA-EPs* substantially prevented SARS-CoV-2 infection in humanized HLA-A*02:01/DR1 and HLA-A*11:01/DR1 transgenic mice. Of note, the immunogenicity of HLA-EPs was not influenced by the multiple mutations across all SARS-CoV-2 variants. Dual immunization with an LNP-formulated mRNA encoding SARS-CoV-2 RBD$_{beta}$ and HLA-EPs improved protection against the SARS-CoV-2 Beta and Omicron BA.1 variants compared with LNP-*RBD$_{beta}$* immunization alone, suggesting an essential role of antigen-specific CD8$^+$ T cells in viral clearance. Indeed, the role of CD8$^+$ T cells in vaccine protection was evaluated by the adenovirus vector-based vaccine Ad26.COV2. S. The study demonstrates that CD8$^+$ T-cells contribute substantially to vaccine protection against SARS-CoV-2 replication, through a CD8-depletion and viral challenge assay[17]. In addition, the combinational protective effect of the humoral and cellular immune responses was analyzed and confirmed by different vaccine strategies, especially for the Omicron variants[52]. Accumulating evidence indicated that an antigen design targeting the cellular immune response or combinational activation of the humoral and cellular immune responses is a promising strategy for developing the next-generation SARS-CoV-2 vaccines.

In this study, we constructed an mRNA format allowing for in situ production of 3 epitope-enriched peptides in tandem (HLA-EPs). A sequence encoding ubiquitin (Ub) with a G76A amino acid modification was placed immediately upstream of the HLA-EPs sequence, which contributes to the rapid formation of the immunopeptidome and efficient access to the HLA compartment of antigen-presenting cells. Indeed, one of the major benefits of mRNA-based vaccines is the endogenous synthesis of encoded proteins, thus allowing the presentation of the foreign antigen by the major histocompatibility complex (MHC)[53]. Nonetheless, the concentration of degraded peptides in the endoplasmic reticulum often acts as a limitation in the MHC complex maturation[54], thus the importance of the proteasome in epitope generation has been broadly accepted[35]. In our mRNA format design, the ubiquitinated peptide is immediately processed to degrade in the proteasome for epitope generation and antigenic presentation, promoting the maturation of cytotoxic T lymphocytes through MHC[55]. Indeed, this strategy with a proteasome targeting peptide has been widely exploited in the construction of T-cell-response-based vaccines against many human viruses[55–57]. In addition, there are other MHC-I trafficking domains, such as MHC class I trafficking domain (MITD)

used by BioNTech[58], for boosting cellular responses to mRNA vaccine. Indeed, the MITD is a sequence located in the cytoplasmic region of MHC class I molecules that controls their recycling between different endolysosomal compartments[59]. The fusion of the MITD trafficking signal with an antigenic peptide can amplify the efficiency of antigen presentation to dendritic cells (DCs). Overall, these fusion peptides with recombinant antigens may improve epitope generation and presentation, thereby enabling the induction of cellular immune responses by genetic vaccines.

The previous study revealed that the T-cell responses in individuals with prior infection and vaccination are largely preserved to Omicron Spike and several nonspike proteins. Nonetheless, a subset of individuals has a more than 50% reduction in T-cell reactivity to the Omicron Spike[22]. Reduced recognition of Omicron spike is primarily observed within the CD8$^+$ T-cell compartment potentially due to escape from HLA binding[22,60]. In our study, we exploited 3 HLA-I epitope-enriched peptides from NSPs to construct an mRNA antigen (HLA-EPs) to stimulate potent cellular immune responses. The 3 selected peptides in the NSPs include many epitopes identified in previous studies and are highly conserved among SARS-CoV-2 VOCs[61–65]. Quite a few epitopes in the HLA-EPs were mutated in the Omicron sublineages; nonetheless, the substitutions in the epitopes did not influence the immunogenicity of epitopes based on bioinformatic analysis, suggesting the advantages of antigen design targeting on SARS-CoV-2 NSPs to develop a T-cell-based vaccine. Therefore, our findings supported to evaluate second-generation vaccine approaches that induce robust T-cell responses targeting both Spike and nonspike antigens to optimize the future design of COVID-19 vaccines.

HLAs are the most polymorphic genes in the human genome. More than 26,000 HLA alleles have been reported in the IMGT/HLA Database (https://www.ebi.ac.uk/ipd/imgt/hla/), with 19,586 class I alleles[66]. Given that the HLA genes exhibit extensive polymorphism, T-cell epitopes are very diverse and heterogeneous among individuals. To ensure that broad T-cell immunity can be effectively activated by immunization with SARS-CoV-2 antigens, we chose 3 regions of the SARS-CoV-2 proteome enriched in human HLA-I epitopes to construct a T-cell-inducing, mRNA-based antigen. These epitopes were predicted to correspond to the 78 most frequent HLA-I alleles, overcoming a great deal of the variability in vaccination efficacy due to HLA heterogeneity among individuals. In contrast to several studies employing peripheral blood mononuclear cells (PBMCs) from convalescent COVID-19 patients[29,67], we identified only a few epitopes in the SARS-CoV-2 N protein with bioinformatic tools. Of note, in our experimental system, the three peptides derived from the nonstructural proteins stimulated much more potent CTL responses than did the HLA epitopes of the M$_{1-113}$ and N$_{Full-length}$ peptides. We, therefore, chose the 3 peptides derived from the nonstructural proteins, rather than peptides derived from the M and N proteins, to construct an mRNA-based vaccine to induce CTL responses to prevent SARS-CoV-2 infection.

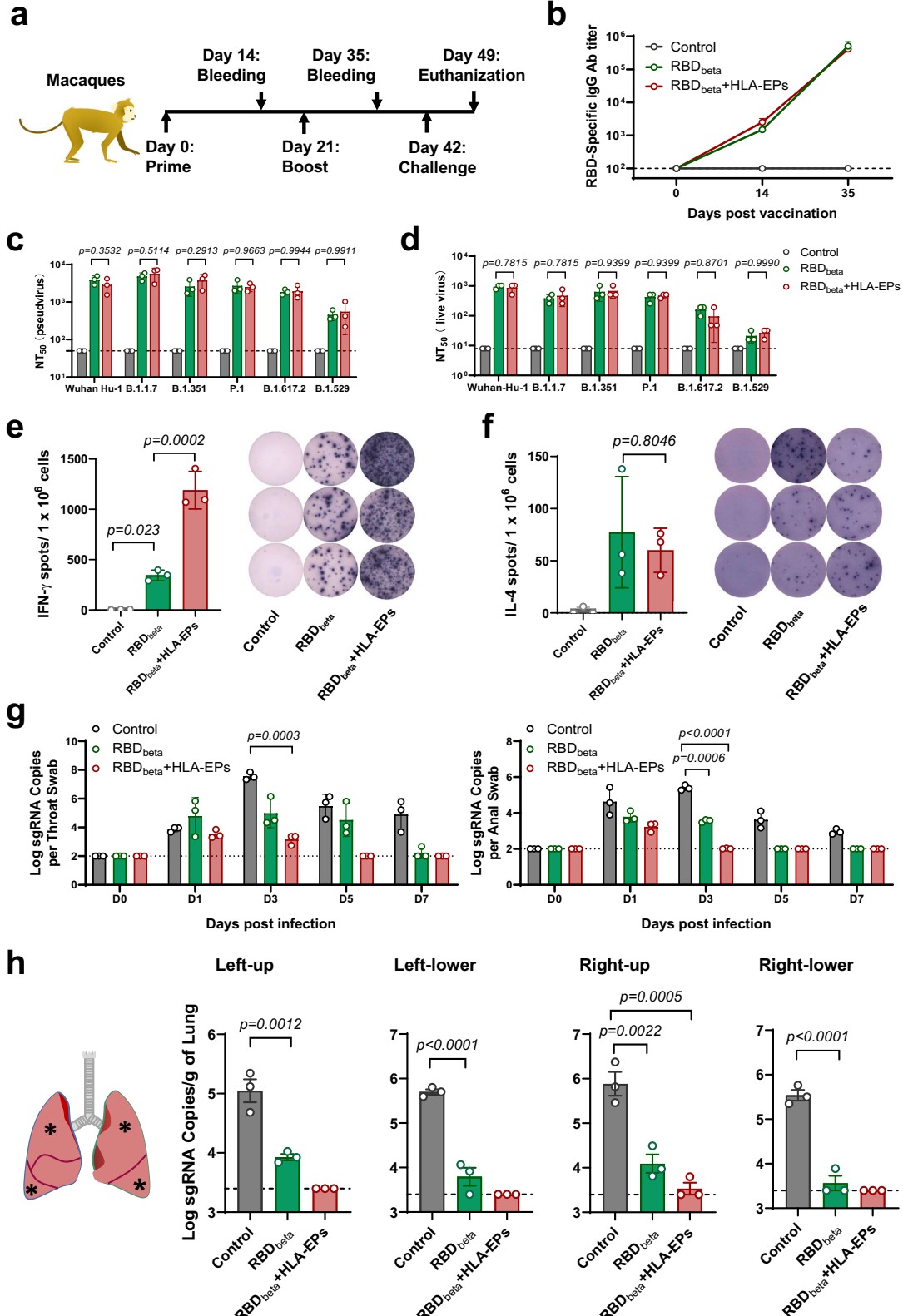

The commercialized COVID-19 vaccines from Pfizer and Moderna used membrane-displayed forms of the Spike proteins. Nonetheless, accumulating evidence indicate that the SARS-CoV-2 RBD have emerged as one of promising antigenic candidates for vaccine design due to the exact target and the potency in stimulating antibodies[5]. Several RBD-based COVID-19 vaccines have been approved for use or entered clinical trials[68]. However, immunization with LNP-*RBD_beta*

mRNA decreased but did not eliminate SARS-CoV-2 infection in the respiratory tissues of *Macaca* monkeys in our experimental settings. Notably, dual immunization with LNP-*HLA-EPs* and LNP-*RBD_beta* rendered improved protection than did LNP-*RBD_beta* vaccination alone in HLA-transgenic mice and nonhuman primates. In future study, we propose to develop an mRNA vaccine combining HLA-EPs with the SARS-CoV-2 full-length Spike, thus might offer a strategy to optimize

**Fig. 4 | Immunological features and protection against SARS-CoV-2 infection conferred by dual immunization with HLA-EPs and RBD antigens in nonhuman primates. a** Schematic diagram of the immunization, sample collection, and challenge schedule. **b–d** The humoral immune response against SARS-CoV-2. SARS-CoV-2 RBD-specific IgG antibodies in the sera of macaques were detected by ELISA using the RBD of the SARS-CoV-2 B.1.351 variant (**b**). The mean ± S.E.M. of RBD-specific IgG titers from 5 biological replicates were plotted against the days post-initial immunization when the sera were collected. The dashed line represents the LOD. The neutralizing antibodies were detected with pseudovirus-based (**c**) and live virus-based (**d**) neutralization assays. Pseudoviruses of six representative variants, Wuhan-Hu-1 (wild-type), B.1.1.7 (Alpha), B.1.351 (Beta), P.1 (Gamma), B.1.617.2 (Delta) and B.1.1.529 (Omicron BA.1), and infectious viruses of the six variants were exploited in the assays. The dashed lines represent the lower limit of quantification of 50 and 8, respectively. **e, f** The Th1-biased cellular immune response. PBMCs collected 14 days post-booster immunization were re-stimulated ex vivo and subjected to IFN-γ ELISpot (LOD = 5) (**e**) and IL-4 ELISpot (LOD = 2) (**f**), and representative dots are shown (right panels). **g, h** The protective efficacy of the vaccine against SARS-CoV-2. Macaques (n = 3) were challenged with $7.0 \times 10^5$ PFU of SARS-CoV-2 (GDPCC-nCoV84, Beta variant) 3 weeks post-booster immunization. The copy number of SARS-CoV-2 sgRNA was determined by qRT-PCR in throat swabs (left panel) and anal swabs (right panel) obtained on Days 0, 1, 3, 5, and 7 post-inoculation (**g**). The LOD was established based on the standard curves at 100 copies per swab. The subgenomic copies in selected lung lobes collected from all macaques 7 days post-inoculation were determined by qRT-PCR (**h**). Data (**c–h**) are shown as the mean ± S.E.M from individual macaques. Source data are provided as a Source Data file. Experiments were repeated twice independently with similar results. Both independent experiments contain 3 technical repeats (**b–h**). Statistical significance was calculated via one-way ANOVA with Tukey's multiple comparison post-hoc two-sided tests (**e, f, h**) or two-way ANOVA with Bonferroni's multiple comparisons (**b–d, g**). P values were adjusted for multiple comparisons.

---

the design of current COVID-19 mRNA vaccines. Overall, our study suggests that optimal COVID-19 vaccines should include broad CTL-inducing and highly conserved SARS-CoV-2 nonstructural protein fragments, as well as neutralizing antibody-inducing antigens.

## Methods

### Ethics statement

Human blood samples were collected with the approval of the local ethics committee at Wenzhou Central Hospital (K2020-01-005 (5)). Human blood collected from donors who provided written informed consent was used for the experiments. All animal care and experimental procedures were approved by the Institutional Animal Care and Use Committee (IACUC) of Shenzhen Bay Laboratory (BACG202101), and National Kunming High-level Biosafety Primate Research Center (DWSP202108 009) in accordance with the relevant guidelines for the protection of animal subjects.

### Cells, viruses, and plasmids

Primary human peripheral blood mononuclear cells (PBMCs) were isolated from convalescent COVID-19 donors, and the basic information about the PBMC donors is summarized in Supplementary Table 2. Isolated PBMCs were cultured in RPMI 1640 medium (Gibco, Cat. No# 11875119) supplemented with 10% (v/v) fetal bovine serum (FBS) (Gibco, Cat. No# 16000-044), 100 units/mL penicillin-streptomycin (Gibco, Cat. No# 15140122), and 60 U/ml IL-2 (Sigma, Cat. No# 11147528001) at 37 °C in 5% $CO_2$. HEK293T cells (Cat. No# CRL-3216) and Vero E6 cells (Cat. No# CRL-1586) were obtained from the American Type Culture Collection (ATCC), and HEK293T cells expressing human ACE2 (HEK293T/hACE2) were kindly provided by Prof. Qiang Ding at Tsinghua University. All cells were cultured in Dulbecco's modified Eagle's medium (DMEM) with GlutaMAX (Gibco, Cat. No# 10566016) supplemented with 10% FBS and 100 units/mL penicillin-streptomycin. SARS-CoV-2 strains (wild-type, B.1.1.7, B.1.351, P.1, B.1.617.2, and B.1.1.529) were cultured and tested in the biosafety level 3 (P3) facilities at the National Kunming High-level Biosafety Primate Research Center in accordance with the relevant guidelines. The plasmid encoding a luciferase reporter expressed from the HIV-1 backbone (pNL4-3.luc.RE) and the plasmid encoding the wild-type SARS-CoV-2 (Wuhan-Hu-1) Spike protein (pcDNA3.1-SARS-2-S) were obtained from Prof. Lu Lu at Fudan University. Point mutations were introduced to the Spike gene, resulting in plasmids encoding the Spike proteins of the SARS-CoV-2 variants that were named pcDNA3.1-B.1.1.7-S, pcDNA3.1-B.1.351-S, pcDNA3.1-P.1-S, pcDNA3.1-B.1.617-S, and pcDNA3.1-B.1.1.529-S. The plasmids pcDNA3.1-NSP-1$_{1-180}$, pcDNA3.1-NSP-3$_{1066-1278}$, pcDNA3.1-NSP-14$_{330-490}$, pcDNA3.1-NSP-3$_{1443-1605}$, pcDNA3.1-NSP-4$_{232-444}$, pcDNA3.1-NSP-6$_{1-201}$, pcDNA3.1-M$_{1-113}$, and pcDNA3.1- N$_{Full-length}$ to express the peptides or protein of interest in the SARS-CoV-2 proteome were constructed and maintained in our laboratory.

### Animals

BALB/c mice, HLA-A*02:01/DR1 transgenic mice, HLA-A*11:01/DR1 transgenic mice, and rhesus macaques were selected for vaccination and/or challenge experiments. All animal care and experimental procedures were approved by the Institutional Animal Care and Use Committee (IACUC) of Shenzhen Bay Laboratory, and National Kunming High-level Biosafety Primate Research Center in accordance with the relevant guidelines for the protection of animal subjects.

### Isolation and expansion of memory CD8⁺ T cells from PBMCs

10 ml of whole blood was gently added to an equal volume of the Ficoll-Paque PLUS (Cytiva, Cat. No# 17-1440-03) layering solution in a 50-mL conical tube at room temperature. The samples were centrifuged at $400 \times g$ for 30 minutes at room temperature. The cloudy phase containing the PBMCs was harvested from the Ficoll-plasma interface, and the PBMCs were washed twice with PBS at $500 \times g$ and 4 °C. Finally, the cells were resuspended for counting and stored in a serum-free cell cryopreservation solution (NCM Biotech, Cat. No# C40100) at −80 °C. Memory CD8⁺ T cells were isolated from the PMBCs using a memory CD8⁺ T-cell isolation kit (Milenyi, Cat. No# 130-094-156) following the manufacturer's instructions. For memory CD8⁺ T-cell expansion, isolated cells were cultured in RPMI medium containing 20% (v/v) FBS, 100 units/mL penicillin, 0.1 mg/mL streptomycin, 0.1 mg/mL Normocin™ (InvivoGen, Cat. No# ant-nr-05), 60 U/ml IL-2, and 25 μL Dynabeads™ Human T-Activator CD3/CD28 (Thermo Fisher Scientific, Cat. No# 11131D) per million cells.

### Analysis of functional SARS-CoV-2 HLA-I epitopes and identification of epitope-enriched fragments

The IEDB server V2.22 (http://tools.iedb.org/tepitool/) and NetMHCpan 4.1 server (http://www.cbs.dtu.dk/services/NetMHCpan/) were used to predict loci of potential HLA-I epitopes for each HLA class I allele listed in Supplementary Table 1 among all structural and nonstructural proteins of the SARS-CoV-2 Wuhan-Hu-1 (wild-type) strain (Accession: NC_045512.2). All available potential epitopes with limiting affinity values (IC50) to <10 nM were selected for further consideration. Epitope-enriched fragments were defined as fragments of SARS-CoV-2 proteins harboring more than 20 such epitopes per 100 amino acids. Fragments of similar lengths harboring few epitopes were also identified. Next, HEK293T cells stably expressing the pHAGE_EF1a_ICADCR, pHAGE_EF1a_IFPGZB, and pHAGE_EF1aHLA vectors (HLA-A*02:01 and HLA-A*11:01 were separately overexpressed in HEK293T cells) were transfected with plasmids encoding the predicted fragments of SARS-CoV-2 ORFs. At 36 h post-transfection, 1 million HEK293T cells were co-cultured with 0.2 million HLA genotype-matched memory CD8⁺ T cells from convalescent COVID-19 patients for 12 h, after which IFP-positive target cells were sorted by flow cytometry (BD LSRFortessa) and analyzed with FlowJo V_10 software.

## Generation of modified mRNA

The mRNAs used in this study were synthesized in vitro using T7 polymerase-mediated DNA-dependent RNA transcription, and the UTPs were completely substituted with 1-methylpseudoUTPs. The incorporated 5′-UTP cap and 3′ poly-A tail modifications were added by the ScriptCap™ Cap 1 Capping System and A-Plus™ Poly(A) Polymerase Tailing Kit for better mRNA stability and translation efficiency. mRNA templates were constructed as such: for $RBD_{beta}$, the mRNA sequences for a signal peptide from tissue plasminogen activator (MDAMKRGLCCVLLLCGAVFVSAS), the receptor-binding domain of the surface protein with three mutant sites, K417N, E484K, and N501Y, from a SARS-CoV-2 strain (P.1.351, PANGO Lineages), and a Flag tag were inserted before the stop codon; for $HLA$-$EPs$, epitope-enriched peptides NSP-$3_{1443-1605}$, NSP-$4_{232-444}$, and NSP-$6_{1-201}$ were linked with A-A-Y motif linkers, with a ubiquitin sequence with a G76A amino modification at the N-terminus and a V5 tag sequence at the C-terminus, and the $HLA$-$NC$ was constructed in the same manner as $HLA$-$EPs$.

## Validation of naked mRNA expression

In vitro expression of naked mRNA in HEK293T cells transfected using TransIT®-mRNA Reagent (Mirus Bio, Cat. No# MIR-2250) was confirmed. Briefly, cells were plated in 6-well plates 24 h prior to transfection, and the medium was changed to Opti-MEM (Gibco, Cat. No# 31985062) 2 h before transfection. One microgram of mRNA was mixed with TransIT®-mRNA Reagent to transfect cells in each well. For RBD$_{beta}$, 72 h later, the supernatants and cells were collected. The supernatant was mixed with 5×SDS loading buffer denatured at 100 °C for 10 min for the Western blot analysis. Antibodies used in the Western blot analysis were an anti-SARS-CoV-2 S1 antibody (Sino Biological, Cat. No# 40150-R007, 1:1000) and a horseradish peroxidase (HRP)-conjugated goat anti-rabbit antibody (Invitrogen, Cat. No# 32460, 1:3000). Collected cells were treated with trypsin-EDTA (Gibco, Cat. No# 25300120) to generate single cells, which were then fixed and permeabilized using Cytofix/Cytoperm reagent (BD Biosciences, Cat. No# BDB554714) according to the manufacturer's instructions. The cells were stained with an anti-Flag tag rabbit mAb (for RBD$_{beta}$, Cell Signaling Technology, Cat. No# 14793 S, 1:1000), followed by a goat anti-rabbit IgG H&L (Alexa Fluor® 555) (Abcam, Cat. No# ab150078, 1:500). The stained cells were analyzed by flow cytometry (BD FACSAria III cell analyzer). Cells treated with the transfection reagent alone were used as controls. For HLA-EPs or HLA-NC, MG132 (APEXBIO, Cat. No# A2585) was added to the cell culture medium 36 h post-transfection. The cells were collected 6 hours later and treated as described above for flow cytometric analyses. The cells were stained by the anti-V5 tag mouse mAb (for HLA-EPs or HLA-NC, Cell Signaling Technology, Cat. No# 80076 S, 1:1000) followed by the goat anti-mouse IgG H&L (Cy3®) (Abcam, Cat. No# ab97035, 1:1000). Mock transfected cells and cells not treated with MG132 were used as two different controls.

## Preparation and characterization of LNP

LNP formulations were prepared using a modified version of a procedure from Precision Nanosystems. Briefly, ionizable lipids were dissolved in ethanol at a volume ratio of 1:1 (25 mM). The lipid mixture was combined with the PNI Formulation Buffer containing $RBD_{beta}$, $HLA$-$EPs$, $HLA$-$NC$, or $Luc$ mRNA at a concentration of 174 µg/mL using microfluidic cartridges. The formulations were concentrated using 10-kDa Amicon Ultra Centrifugal Filters, passed through a 0.22-µm filter, and stored at 4 °C until use. The concentrations of all formulations defined as LNP-$RBD_{beta}$, LNP-$HLA$-$EPs$, and LNP-$NC$ were tested using the Quant-iT RiboGreen RNA Kit (Invitrogen, Cat. No# R11490). The particle size distribution of LNPs was analyzed by dynamic light scattering (Dynapro NanoStar).

## Assessment of the in vivo distribution of mRNA-LNPs

To detect the in vivo distribution of mRNA-LNPs, female BALB/c mice aged 6–8 weeks ($n = 3$) were inoculated with 2 µg of Luciferase mRNA-LNP via intramuscular (i.m.) routes. The animals were injected intra-peritoneally (i.p.) with D-Luciferin (Solarbio, Cat. No# D8390) 6 h post-inoculation. The equal volume PBS injected mice were set as control ($n = 3$). After a 3-minute incubation, fluorescence signals were examined with an IVIS Spectrum instrument (Lumina II) for 60 s. For further in vitro imaging, tissues including the brain, heart, liver, spleen, lungs, kidneys, muscles, and lymph nodes were collected immediately, and the fluorescence signals of each tissue were collected with the IVIS imager for 60 s.

## Animal immunization

**Mice. a.** HLA-A*02:01/DR1 or HLA-A*11:01/DR1 transgenic mice (male and female, 4–5 weeks) were randomly allocated into three groups ($n = 15$). LNP-$HLA$-$EPs$ diluted in PBS with 8.7% sucrose were injected intramuscularly (i.m.) following a prime and boost regimen with an interval of 3 weeks (0.5 or 10 µg per dose), and LNP-$NC$ (10 µg per dose) served as the control. The immunized mice were further processed for subsequent experiments as described below. **b.** HLA-A*02:01/DR1 transgenic mice (4–5 weeks) were randomly allocated into three groups ($n = 20$) and vaccinated via the intramuscular route with the following agents: LNP-$RBD_{beta}$ (0.5 µg per mouse), LNP-$RBD_{beta}$ + LNP-$HLA$-$EPs$ (0.5 µg per mouse for both components), or PBS with 8.7% sucrose as buffer controls. The same doses were used for a booster immunization at 3 weeks post-priming vaccination. Five mice per group were sacrificed at 7 days post-booster vaccination to analyze germinal center B (GC B) cells, follicular helper T (Tfh) cells, IFN-γ, IL-4, CXCR13, and CXCR5 as described below. Serum samples were collected 3 days before the priming vaccination, 7 and 21 days after the boost vaccination, and analyzed as described below. The rest mice were further processed for subsequent challenge experiments as described below.

**Macaques.** Female rhesus macaques (2–3 years old) were randomly assigned to three groups ($n = 3$): LNP-$RBD_{beta}$ (100 µg per macaque), LNP-$RBD_{beta}$ + LNP-$HLA$-$EPs$ (100 µg per macaque for both components), and PBS with 8.7% sucrose. On days 0 and 21, vaccines were administered intramuscularly in the left deltoid muscle. The macaques were anesthetized with ketamine HCl (10 mg/kg; intramuscularly) prior to immunization and monitored during sedation. Blood was collected 3 days before vaccination and 14 days after each injection to isolate the serum and PBMCs. The immunized macaques were further processed for subsequent challenge experiments as described below.

## ELISA analysis of serum RBD-specific IgG

To detect SARS-CoV-2 RBD-specific antibodies in immunized sera collected from mice or macaques, plates were coated with SARS-CoV-2 RBD proteins (Sino Biological: wild-type, Cat. No# 40592-V08H; Alpha strain, Cat. No# 40592-V08H82; Beta strain, Cat. No# 40592-V08H84; Gamma strain, Cat. No# 40592-V08H86; Delta strain, Cat. No# 40592-V08H91; Omicron strain, Cat. No# 40592-V08H121) at a concentration of 1 µg/ml, followed by sequential addition of serially diluted serum samples and HRP-conjugated anti-mouse IgG (1:5000) antibodies (Invitrogen, Cat. No# A16066) or HRP-conjugated anti-monkey IgG (1:5000) antibodies (Invitrogen, Cat. No# PA1-84631) for 1 h at 37 °C. The plates were sequentially incubated with the substrate TMB (3,3′,5,5′-tetramethylbenzidine) (Sigma, Cat. No# T0440) and then H$_2$SO$_4$ (1 N) to stop the reaction. The absorbance at 450 nm was measured on a microplate reader.

## Neutralization of live SARS-CoV-2

To assess the neutralizing activity of immunized serum samples against live SARS-CoV-2, a cytopathic effect (CPE)-based microneutralization

assay was used[5]. Briefly, Vero E6 cells ($5 \times 10^4$ cells) were seeded in 96-well plates and cultured at 37 °C to form a monolayer. Circulating SARS-CoV-2 strains (wild-type, B.1.1.7, B.1.1.351, P.1, B.1.627.2, and B.1.1.529) at the 50% tissue-culture infectious dose ($TCID_{50}$) were thoroughly mixed with an equal volume of diluted sera from immunized mice or macaques before addition to the cells. After incubating at 37 °C for 1 h, the serum-virus mixture was added to the Vero E6 cells. On Day 3 after infection, cytopathogenic effects were recorded under a microscope, and the neutralizing titers were calculated from the serial dilutions of sera that resulted in EC50 inhibition.

### Generation of SARS-CoV-2 pseudoviruses and a pseudovirus neutralization assay

SARS-CoV-2 pseudoviruses were generated, and pseudovirus neutralization assays were performed[69]. In brief, HEK293T cells were cotransfected with a plasmid encoding an Env-defective, luciferase-expressing HIV-1 genome (pNL4-3.luc. RE) and one of the plasmids encoding the codon-optimized S protein of a circulating SARS-CoV-2 strain. Six to eight hours later, the medium was replaced with fresh complete medium. The pseudovirus-containing supernatants were collected 48 h after transfection for titration on target cells. SARS-CoV-2 pseudoviruses were incubated with serially diluted serum at 37 °C for 1 h and added to HEK293T/hACE2 cells, followed by the addition of fresh medium 24 h later. The cells were lysed 72 h later in cell lysis buffer (Promega, Cat. No# E194A), incubated with a luciferase substrate (Promega, Cat. No# E1910), and assessed for relative luciferase activity using a Varioskan Flash microplate reader (Thermo Scientific). The 50% SARS-CoV-2 pseudovirus neutralizing antibody titer ($NT_{50}$) was calculated to define the neutralization potency.

### ELISpot assays of IFN-γ and IL-4

**Mice.** ELISpot assays were performed with mouse IFN-γ (Abcam, Cat. No# ab64029) and IL-4 ELISpot kits (Mabtech, Cat. No# 3311-4HPW-10) according to the manufacturer's instructions. Counted splenocytes were restimulated ex vivo with the $RBD_{beta}$ 15-amino-acid overlapping peptide pool (for the LNP-$RBD_{beta}$ group, 2 µg/ml per peptide), $RBD_{beta}$ and HLA-EPs 15-amino-acid overlapping peptide pool mixture (for the LNP-$RBD_{beta}$ + LNP-$HLA$-$EPs$ group, 2 µg/ml per peptide), or the DMSO control.

**Macaques.** ELISpot assays were performed with nonhuman primate IFN-γ (Mabtech, Cat. No# 3421M-4APT-10) and IL-4 ELISpot kits (Mabtech, Cat. No# 3410-2APW-10) according to the manufacturer's instructions. A total of $2 \times 10^5$ PBMCs (IFN-γ) or $5 \times 10^5$ PBMCs (IL-4) were stimulated ex vivo with the $RBD_{beta}$ 15-amino-acid overlapping peptide pool (for the LNP-$RBD_{beta}$ group, 2 µg/ml per peptide), $RBD_{beta}$ and HLA-EPs 15-amino-acid overlapping peptide pool (for the LNP-$RBD_{beta}$ + LNP-$HLA$-$EPs$ group, 2 µg/ml per peptide), or the DMSO control. Spots were counted using an ELISpot reader (iSpot).

### Profiling of cells in the lymph nodes and splenocytes

Inguinal lymph nodes from immunized mice ($n = 5$, HLA-A*02:01/DR1 transgenic mouse under regimen **b**.) were harvested at day 7 post-second immunization and pooled. The nodes were homogenized into single-cell suspensions using a syringe plunger and passed through a 70 µm cell strainer in complete RPMI 1640 media containing 10% fetal bovine serum. Cells were washed and resuspended in fresh RPMI-10% FBS media in a 96-well round-bottomed plate for staining. Cells were first stained with Ghost Dye™ Red 780 (TONBO Biosciences, Cat. No# 13-0865-T100) for dead cells, and then stained with a cocktail of the following fluorescently labeled antibodies: anti-CD45-Alexa Fluor™ 700 (Invitrogen, Cat. No# 56-0451-82), anti-CD4-FITC (Tonbo Biosciences, Cat. No# 35-0042-U100), anti-CD185-Brilliant Violet 605™ (BioLegend, Cat. No# 145513), anti-PD-1-Brilliant Violet 421™ (BioLegend, Cat. No# 135218), anti-B220-PerCP-Cyanine5.5 (TONBO

Biosciences, Cat. No# 65-0452-U100), anti-CD95-PE (BioLegend, Cat. No# 152608), anti-GL-7-APC (BioLegend, Cat. No# 144618) in the cell staining buffer (BioLegend, Cat. No# 420201) and incubated for 20 min in the dark at room temperature. Cells were then washed and resuspended in cell staining buffer and were acquired using BD Aria III cell analyzer, and data were analyzed using FlowJo software V_10.

The spleens from immunized mice ($n = 5$, HLA-A*02:01/DR1 transgenic mouse under regimen **a**. and **b**. or HLA-A*11:01/DR1 transgenic mouse under regimen **a**.) were harvested on Day 21 post-second immunization. The spleens were briefly lysed and suspended in ammonium-chloride-potassium buffer to lyse red blood cells. The splenocytes were then washed and resuspended in RPMI medium 1640 supplemented with 10% fetal bovine serum. Cells were first stained with Ghost Dye™ Red 780 (TONBO Biosciences, Cat. No# 13-0865-T100) for dead cells, and then stained with a cocktail of the following fluorescently labeled antibodies: anti-CD45-Alexa Fluor™ 700 (Invitrogen, Cat. No# 56-0451-82), anti-CD4-FITC (Tonbo Biosciences, Cat. No# 35-0042-U100), anti-CD8-PerCP-Cyanine5.5 (TONBO Biosciences, Cat. No# 65-0081-U100), anti-CD44-APC (BioLegend, Cat. No# 103012), anti-CD62L-BV421 (BioLegend, Cat. No# 104436) in the cell stain buffer (BioLegend, Cat. No# 420201) and incubated for 20 min in the dark at room temperature. For the intracellular cytokine staining (ICS), the cells underwent additional fixation and permeabilization using the Cytofix/Cytoperm reagent (BD Biosciences, Cat. No# BDB554714) according to the manufacturer's instructions and staining with the anti-IFN-γ-PE (TONBO Biosciences, Cat. No# 50-7311-U100) and/or anti-TNF-α-BV421 (BioLegend, Cat. No# 506328) antibody. Cells were then washed and resuspended in cell stain buffer and were acquired using BD Aria III cell analyzer and data were analyzed using FlowJo software V_10.

### CXCR13 and CXCR5 expression in the lymph nodes

The inguinal lymph nodes of immunized mice (male and female, $n = 5$, HLA-A*02:01/DR1 transgenic mice) were harvested on Day 7 after the second immunization. RNA extracted from the lymph nodes was subjected to CXCR13 and CXCR5 mRNA level measurement by qRT-PCR. The primer sequences are shown in Supplementary Table 5. The expression levels of CXCR13 and CXCR5 were normalized to those of mouse GAPDH (GenBank number: NM_001411843.1).

### HLA tetramer preparation and staining assay

Predicted functional wild-type peptides, associated mutant peptides, and reported positive control peptides (Supplementary Fig. 3 and Supplementary Table 4) matched with HLA-A*02:01 or HLA-A*11:01 were synthesized and used for peptide-specific tetramer preparation following the instructions of the HLA-A*02:01 Tetramer Kit-PE (MBL International Corporation, Cat. No# TB-7300-K1) or HLA-A*11:01 Tetramer Kit-PE (MBL International Corporation, Cat. No# TB-7304-K1)[32,48,70,71]. Splenocytes isolated from LNP-$HLA$-$EPs$- or LNP-$NC$-immunized HLA-A*02:01/DR1 transgenic mice or HLA-A*11:01/DR1 transgenic mice were stimulated with the above peptides at 2 µg/ml per peptide. After stimulation cells were washed and stained with Ghost Dye™ Red 780 (TONBO Biosciences, Cat. No# 13-0865-T100) for dead cells. Surface markers were stained with CD45-Alexa Fluor™ 700 (Invitrogen, Cat. No# 56-0451-82), CD8-PerCP-Cyanine5.5 (TONBO Biosciences, Cat. No# 65-0081-U100) and Peptide-tetramer-PE mixture, for 20 min at room temperature. The stained cells were pelleted and washed three times before being analyzed by flow cytometry (BD Aria III cell analyzer).

### Profiling of macaque cytokines by ELISA

Immunized macaque PBMCs were restimulated for 48 h with the appropriate 15-amino-acid overlapping peptide pools (0.2 µg/ml per peptide) or cell culture medium (no peptides) as a control. The concentrations of IFN-γ, TNF-α, IL-4, and IL-5 in the supernatants were

determined using ELISA kits (Thermo Fisher Scientific) according to the manufacturer's instructions. The OD at 450 nm was measured with a Varioskan Flash microplate reader (Thermo Scientific).

## SARS-CoV-2 challenge studies

**Mice.** Immunized mice were challenged on Day 21 post-booster immunization in the following experiments to evaluate vaccine efficacy. (1). LNP-*HLA-EPs* (0.5 μg and 10 μg per dose) or LNP-*NC* (10 μg) immunized HLA-A*02:01/DR1 transgenic mice or HLA-A*11:01/DR1 transgenic mice were injected intraperitoneally with an anti-mouse anti-CD8-α (IgG2b) mAb (Bio X Cell, Cat. No# BE0061, 100 μg/mouse, $n = 5$) or IgG2b isotype control mAb (Bio X Cell, Cat. No# BE0090, 100 μg/mouse, $n = 5$) on Days −1 and 0 of the challenge. Peripheral blood cells were evaluated for CD8⁺ T-cell depletion by flow cytometric analysis on the day of the challenge. The mice were inoculated intranasally with SARS-CoV-2 (GDPCC-nCoV84, South African strain, Beta variant, PANGO lineage B.1.351, $1.4 \times 10^5$ PFU) and sacrificed at 4 days postinfection. The lung and trachea tissues were collected for viral load detection by a plaque assay and qRT-PCR. (2) HLA-A*02:01/DR1 transgenic mice immunized with LNP-$RBD_{beta}$ (0.5 μg per dose), LNP-$RBD_{beta}$ (0.5 μg per dose)+LNP-*HLA-EPs* (0.5 μg per dose), or PBS with 8.7% sucrose were challenged with SARS-CoV-2 ($1.4 \times 10^5$ PFU Beta strain or $2.0 \times 10^5$ PFU Omicron strain) at 21 days post-boost immunization using the same procedure. The viral subgenomic RNA was quantified with qRT-PCR using FastKing One Step Probe RT-qPCR kit (Tiangen Biotech Cat. No# FP314-01) and a Bio-Rad CFX-96 Touch Real-Time Detection System. The primer and probe sequences used were derived from the E gene and are listed in Supplementary Table 5. For absolute quantification, standard curves were generated using serial dilutions of a cDNA plasmid of known concentration, and the limit of detection was set at $2.5 \times 10^3$ copies per gram tissue.

**Macaques.** Three groups of immunized adult nonhuman primates ($n = 3$) were used for the challenge study performed with live SARS-CoV-2. On Day 21 after the second vaccination, macaques were challenged with $7.0 \times 10^5$ PFU of the SARS-CoV-2 Beta variant in a total volume of 1.0 ml, split between the trachea (0.5 ml each) and nares (0.5 ml each). Throat and anal swabs were collected 0, 1, 3, 5, and 7 days post-inoculation. The macaques were euthanized, and lung tissues were collected 7 days post-inoculation. The viral subgenomic RNA was quantified with qRT-PCR in lung tissues, throat swabs, and anal swabs. The primer and probe sequences used were derived from the E gene and are listed in Supplementary Table 5. For absolute quantification, standard curves were generated using serial dilutions of a cDNA plasmid of known concentration, and the limit of detection was set at $2.5 \times 10^3$ copies per gram tissue, or 100 copies per swab.

## Histopathological assay

Lung tissues from mice or macaques were collected immediately, fixed in 10% neutral-buffered formalin without inflation, and embedded in paraffin. Approximately 5-μm sections were cut and mounted on slides. Histopathological changes caused by SARS-CoV-2 infection were examined by standard H&E staining and viewed under a light microscope. H&E-stained lung tissue sections were blindly examined and scored by trained histo-pathologists.

## Statistics

Statistical analyses were performed using Prism 8.0 (GraphPad software). Comparisons among multiple groups were performed using one-way ANOVA followed by Tukey's multiple comparison post hoc two-sided tests or two-way ANOVA followed by Bonferroni's multiple comparison post hoc two-sided tests. Detailed statistical descriptions were provided in the figure legends.

## Reporting summary

Further information on research design is available in the Nature Portfolio Reporting Summary linked to this article.

## Data availability

The authors declare that all data supporting the results of this study are available in the paper and supplementary information. Source data are provided in this paper.

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

## Acknowledgements

This work was funded by grants from the National Key Research and Development Plan of China (2022YFC2303400, 2021YFC2300200, 2018YFA0507202, 2020YFC1200104, 2021YFC2302405 and 2022YFC2303200). The National Natural Science Foundation of China (32188101, 31825001, and 81961160737) to G.C. The National Natural Science Foundation of China (82271872 and 32100755) to W.T. The Emergency Key Program of Guangzhou Laboratory (EKPG21-33) to G.C. Innovation Team Project of Yunnan Science and Technology Department (202105AE160020) to G.C. Tsinghua-Foshan Innovation Special Fund (TFISF) (2022THFS6124) to G.C. Shenzhen San-Ming Project for prevention and research on vector-borne diseases to G.C.

## Author contributions

G.C. designed and supervised the study, and wrote and revised the manuscript. W.T., S.L., G.Z., X.Z., W.Y., D.C, S.F., B.C., L.R., and H.S. conducted the study and analyzed the data. Z.C. and M.P. helped with the animal experiment. Q.S. and X.P. supervised the work in BSL-3. J.Lu prepared the mRNA antigens by the lipid nanoparticle (LNP)-formulated nucleoside-modified mRNA technology. Q.S., X.P., J.L., X.T., and P.W. contributed experimental design and strengthened the writing of the manuscript. All authors reviewed, critiqued, and provided comments on the text.

## Competing interests

The authors declare no competing interests.

## Additional information

¹Institute of Infectious Diseases, Shenzhen Bay Laboratory, Shenzhen 518132, China. ²Tsinghua-Peking Joint Center for Life Sciences, School of Medicine, Tsinghua University, Beijing 100084, China. ³State Key Laboratory of Respiratory Disease, National Clinical Research Center for Respiratory Disease, Guangzhou Institute of Respiratory Health, First Affiliated Hospital of Guangzhou Medical University, Guangzhou 510182, China. ⁴National Kunming High-level Biosafety Primate Research Center, Institute of Medical Biology, Chinese Academy of Medical Sciences and Peking Union Medical College, Kunming 650118, China. ⁵State Key Laboratory of Pathogen and Biosecurity, Beijing Institute of Microbiology and Epidemiology, Academy of Military Medical Sciences, Beijing 100071, China. ⁶Zhejiang Provincial Key Laboratory of Medical Genetics, Key Laboratory of Laboratory Medicine, Ministry of Education, School of Laboratory Medicine and Life Sciences, Wenzhou Medical University, Wenzhou 325035, China. ⁷Wenzhou Central Hospital, Wenzhou 325000, China. ⁸Beijing Advanced Innovation Center for Structural Biology, Beijing Frontier Research Center for Biological Structure, MOE Key Laboratory of Bioorganic Phosphorus Chemistry & Chemical Biology, School of Pharmaceutical Sciences, Tsinghua University, Beijing 100084, China. ⁹State Key Laboratory of Genetic Engineering, School of Life Sciences, Zhongshan Hospital, Shanghai Institute of Infectious Disease and Biosecurity, Fudan University, Shanghai 200438, China. ¹⁰Department of Immunology, School of Medicine, the University of Connecticut Health Center, Farmington, CT 06030, USA. ¹¹These authors contributed equally: Wanbo Tai, Shengyong Feng, Benjie Chai, Shuaiyao Lu, Guangyu Zhao, Dong Chen, Wenhai Yu. ✉e-mail: linjinzhong@fudan.edu.cn; qsun@imbcams.com.cn; pengxiaozhong@pumc.edu.cn; gongcheng@mail.tsinghua.edu.cn

