## [Peer Review File · Nature Communications]

An mRNA-based T-cell-inducing antigen strengthens COVID-19 vaccine against SARS-CoV-2 variantsEditorial Note: This manuscript has been previously reviewed at another journal that is not operating a transparent peer review scheme. This document only contains reviewer comments and rebuttal letters for versions considered at Nature Communications.

Reviewers' Comments:

Reviewer #1:

Remarks to the Author:

The manuscript entitled "An mRNA-based T-cell-inducing antigen strengthens COVID-19 vaccine against SARS-CoV-2 variants" by Tai et al. has identified HLA-I epitope-enriched peptides which can activate CD8+ T lymphocytes from convalescent COVID-19 patients. These peptides, together with a proteasome targeting peptide used to enhance MHC-I presentation, were then formulated into the vaccine prototype HLA-EPs mRNA. The robustness of the epitope selection pipeline was successfully demonstrated using three SARS-CoV-2 peptides with low scores from the in-silico analyses. The authors then observed immunogenicity induced by the HLA-EPs in both HLA-A*02:01/DR1 and HLA-A*11:01/DR1 transgenic mice. In addition, they proposed that dual immunization with HLA-EPs and RBD in mice and in non-human primates could protect the animals from viral challenges. This work presents an interesting SARS-CoV-2 biology study in which T cell vaccines can eventually be exploited in clinical trials. This paper is potentially interesting for the field of COVID-19 immunology and vaccinology. The current manuscript is ready in its present form to publish by the journal.

Reviewer #4:

Remarks to the Author:

The authors have addressed my concerns.

Reviewer #5:

Remarks to the Author:

The authors have satisfactorily addressed the concerns raised by Reviewer #2.

However, please note that many of the flow plots and corresponding axes titles in the supplementary data need attention.

Reviewer #1 (Remarks to the Author):

*The manuscript entitled “An mRNA-based T-cell-inducing antigen strengthens COVID-19 vaccine against SARS-CoV-2 variants” by Tai et al. has identified HLA-I epitope-enriched peptides which can activate CD8+ T lymphocytes from convalescent COVID-19 patients. These peptides, together with a proteasome targeting peptide used to enhance MHC-I presentation, were then formulated into the vaccine prototype HLA-EPs mRNA. The robustness of the epitope selection pipeline was successfully demonstrated using three SARS-CoV-2 peptides with low scores from the in-silico analyses. The authors then observed immunogenicity induced by the HLA-EPs in both HLA-A*02:01/DR1 and HLA-A*11:01/DR1 transgenic mice. In addition, they proposed that dual immunization with HLA-EPs and RBD in mice and in non-human primates could protect the animals from viral challenges. This work presents an interesting SARS-CoV-2 biology study in which T cell vaccines can eventually be exploited in clinical trials. This paper is potentially interesting for the field of COVID-19 immunology and vaccinology. The current manuscript is ready in its present form to publish by the journal.*

Responses to Reviewer #1: We appreciate reviewer’s positive comments and approval.

Reviewer #4 (Remarks to the Author):

The authors have addressed my concerns.

Responses to Reviewer #4: We appreciate reviewer’s approval.

Reviewer #5 (Remarks to the Author):

The authors have satisfactorily addressed the concerns raised by Reviewer #2.

However, please note that many of the flow plots and corresponding axes titles in the supplementary data need attention.

Responses to Reviewer #5: Thanks, We checked and revised the flow plots and corresponding axes titles in the supplementary data.